# ANTICIPATING ETHICAL ISSUES IN IMAGE TRAINING-DATA USING LARGE-SCALE TRANSFORMERS

## ABSTRACT

⚠ This paper contains images and descriptions that are offensive in nature.
Probing or fine-tuning (large-scale) pre-trained models results in state-of-the-art performance for many NLP tasks and, more recently, even for computer vision tasks when combined with image data. Unfortunately, these approaches also entail severe risks. In particular, large image datasets automatically scraped from the web may contain derogatory terms as categories and offensive images, and may also underrepresent specific classes. Consequently, there is an urgent need to carefully document datasets and curate their content. Unfortunately, this process is tedious and error-prone. We show that pre-trained transformers themselves provide a methodology for the automated curation of large-scale vision datasets. Based on human-annotated examples and the implicit knowledge of a CLIP based model, we demonstrate that one can select relevant prompts for rating the offensiveness of an image. In addition to e.g. privacy violation and pornographic content previously identified in ImageNet, we demonstrate that our approach identifies further inappropriate and potentially offensive content.

## 1 INTRODUCTION

Deep learning models yielded many improvements in several fields. Particularly, transfer learning from models pre-trained on large-scale supervised data has become common practice in many tasks both with and without sufficient data to train deep learning models. Whereas approaches like semi-supervised sequence learning (Dai & Le, 2015) and datasets such as ImageNet (Deng et al., 2009), especially the ImageNet-ILSVRC-2012 dataset with 1.2 million images, established pre-training approaches, in the following years, the training data size increased rapidly to billions of training examples (Brown et al., 2020; Jia et al., 2021), steadily improving the capabilities of deep models. Recently, autoregressive (Radford et al., 2019), masked language modeling (Devlin et al., 2019) as well as natural language guided vision models (Radford et al., 2021) have enabled zero-shot transfer to downstream datasets removing the need for dataset-specific customization. Besides the parameter size of these models, the immense size of training data has enabled deep learning models to achieve high accuracy on specific benchmarks in natural language processing (NLP) and computer vision (CV) applications. However, in both application areas, the training data has been shown to have problematic characteristics resulting in models that encode *e.g.* stereotypical and derogatory associations (Gebru et al., 2018; Bender et al., 2021). Unfortunately, the curation of these large datasets is tedious and error-prone. Pre-trained models (PM) used for downstream tasks such as face detection propagate retained knowledge to the downstream module *e.g.* the classifier.

To raise the awareness of such issues, Gebru et al. (2018) described how large, uncurated, Internet-based datasets encode *e.g.* dominant and hegemonic views, which further harms people at the margins. The authors urge researchers and dataset creators to invest significant resource allocation towards dataset curation and documentation practices. As a result, Birhane & Prabhu (2021) provided modules to detect faces and post-process them to provide privacy, as well as a pornographic content classifier to remove inappropriate images. Furthermore, Birhane & Prabhu (2021) conducted a hand surveyed image selection to identify misogynistic images in the ImageNet-ILSVRC-2012 dataset. Unfortunately, such a curation process is tedious and does not scale to current dataset sizes. Moreover, misogynistic images, as well as pornographic content, are only two subsets of offensive images. It remains an open question how to infer general offensiveness in images, including abusive, indecent, obscene, or menacing content, and how to identify them in an automated dataset curation process.

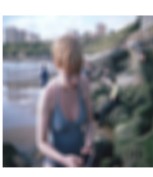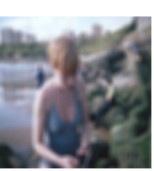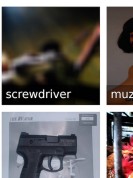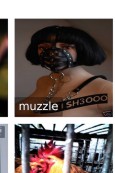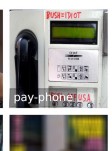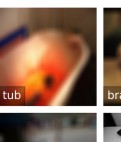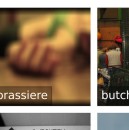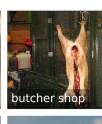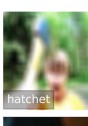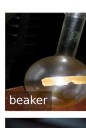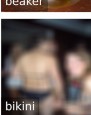

Figure 1: Results from the ImageNet-ILSVRC-2012 dataset (validation set). Left: The single image identified by the hand surveyed image selection of Birhane & Prabhu (2021). Right: Range of samples from our CLIP pre-selection. In summary, CLIP is detecting over $1,5k$ out of $50,000$ images from ImageNet's validation set as possible offending and over $30k$ out of $1,281,167$ from the training set. Like our classifier, the pornographic classifier used in (Birhane & Prabhu, 2021) identifies the 5th image in the first row as inappropriate. However, our classifier finds additional images along other dimensions of inappropriate content. This provides an extension to private obfuscation of the faces and pornographic content classifiers provided by Birhane & Prabhu (2021). We blurred the images to not offend the reader and to not violate privacy.

Whereas large image datasets automatically scraped from the web may contain derogatory terms as categories and offensive images, which results in models with undesirable behavior, pre-trained models may also reflect desirable implicit knowledge and biases such as our social, ethical, and moral choices (Jentzsch et al., 2019; Schramowski et al., 2020) reflected within the training data.

In our study, we investigate modern vision PMs trained on large-scale datasets, in particular, the Contrastive Language-Image Pre-trained model (CLIP) (Radford et al., 2021) and argue that they themselves pave a way to mitigate the associated risks. Specifically, we show that they encode implicit knowledge to infer offensiveness in images overcoming previous issues, namely the lack of adequate and sufficient training data. Furthermore, we demonstrate that our approach can be utilized to annotate offensive images in vision datasets and, therefore, reliably assist the curation process of such datasets. We illustrate our approach on the popular ImageNet-ILSVRC-2012 dataset and show that large computer vision datasets contain additional inappropriate content, which previous documentations had not detected. With our proposed method this content can be automatically and reliably pre-selected.

As an example, Fig. 1(left) shows an exemplary image from the ImageNet-ILSVRC-2012 validation set identified as misogynistic content —categorized as "beach voyeur"— by a hand-surveyed image selection in (Birhane & Prabhu, 2021). Next to this human-selected image, Birhane & Prabhu (2021) applied different models to detect visible faces (thus violating privacy rights) and pornographic content. However, as we will show with our study, further inappropriate images, which we refer to as offensive, can be identified within the dataset. For instance, Fig. 1(right) shows sixteen hand-picked images from a set of automatically detected possibly offensive images, utilizing our proposed approach. Depending on the task and stakeholders, this ranges from offensive objects such as weapons (second row, first and fifth image) and dead animals (first row, sixth image) to immoral actions such as harming or even killing animals (second row, second image) and humans (second row, seventh image), as well as offensive text and symbols (first row, third image).

With our study we therefore strongly advocate for curating and documenting a dataset by the categories and models provided by Birhane & Prabhu (2021) but also by taking the possible general offensiveness in images into account. To this end, we provide our models and the necessary data to reproduce our experiments and utilize our proposed method[1].

We proceed as follows. We start with a brief overview of related work and required background introducing pre-trained models and their successes as well as concerns raised. Next, we describe the term offensiveness and show that common deep models can not reliably detect offensive image content due to the lack of sufficient data. We then continue by demonstrating that recent models, guided by natural language during the pre-training phase, can infer offensiveness in images based on their implicit knowledge. Before concluding, we present our automated dataset curation exemplary on the ImageNet-ILSVRC-2012 dataset.

---

[1]Anonymous link placeholder. Please see supplement.

## 2 BACKGROUND AND RELATED WORK

**Concerns about large-scale data sets.** Pre-training has become an essential approach in many vision and language tasks. In the vision domain, pre-training on large-scale supervised data such as ImageNet (Deng et al., 2009) has shown to be crucial for enhancing performance on downstream tasks via transfer learning. Since these datasets contain millions of data samples, curating such pre-training datasets requires heavy work on data gathering, sampling, and human annotation, making it error-prone and difficult to scale. Moreover, in the language domain, task-agnostic objectives such as autoregressive (Radford et al., 2019) and masked language modeling (Devlin et al., 2019) have scaled across many orders of magnitude, especially in model capacity and data, steadily increasing performance but also the capabilities of deep models. With their standardized input-output (text-to-text) interface Radford et al. (2019) have enabled zero-shot transfer to downstream datasets. Recent systems like GPT-3 (Brown et al., 2020) are now competitive across many tasks with specialized models whereas requiring only a small amount to no task-specific training data. Based on these advances, more recently, Radford et al. (2021) and Jia et al. (2021) introduced models with similar capabilities in the vision domain. However, pre-training such models requires particularly large-scale training data, and the datasets' curation process is tedious and error-prone.

To tackle this issue Gebru et al. (2018) suggest to provide dataset audit cards to document datasets. This provides stakeholders the ability to understand training data characteristics in order to alleviate known as well as unknown issues. The authors argue that whereas documentation allows for potential accountability, undocumented training data perpetuates harm without recourse.

Birhane & Prabhu (2021) provided such a dataset card for the popular computer-vision ImageNet-ILSVRC-2012 dataset, including several metrics and the hand surveyed identification of images with misogynistic content. More importantly, the authors raised the awareness of polluted image datasets by the example of pornographic content inside several popular computer vision benchmark datasets. Although the authors raised criticism against ImageNet and identified several inappropriate images, the ImageNet-ILSVRC-2012 dataset —and the pre-trained models— are still under the most popular datasets in the ML community. In line with Gebru et al. (2018), Birhane & Prabhu (2021) urge that ethics checks for future dataset curation endeavors become an integral part of the human-in-the-loop validation phase.

**ImageNet.** The ImageNet (Deng et al., 2009) data collection is one of the most popular datasets in the computer vision domain and mostly refers to the subset ImageNet1k dataset with 1.2 million images across 1000 classes. This was introduced in 2012 for the classification challenge in the ImageNet Large Scale Visual Recognition Challenge (ILSVRC). However, in total the collection (ImageNet21k) covers over 14 million images spread across 21,841 classes.

As Birhane & Prabhu (2021) state, the ImageNet dataset remains one of the most influential and powerful image databases available today, although it was created over a decade ago. To apply transfer learning, the most popular deep learning frameworks provide downloadable pre-trained models for ImageNet1k. Recently, Ridnik et al. (2021) provided a novel scheme for high-quality, efficient pre-training on ImageNet21k and, along with it, the resulting pre-trained models.

**Pre-training vision models with natural language supervision.** Pre-training methods that learn directly from raw data have revolutionized many tasks in natural language processing and computer vision over the last few years. Radford et al. (2021) propose visual representation learning via natural language supervision in a contrastive learning setting. The authors collected over 400M image-text pairs (WebImageText dataset) to show that the improvement with large-scale transformer models in NLP can be transferred to vision. More precisely, whereas typical vision models jointly train an image feature extractor and a linear classifier, CLIP jointly trains an image encoder and a text encoder to predict the correct pairings of a batch of (image, text) training examples. At test time the authors propose to synthesize the learned text encoder with a (zero-shot) linear classifier by embedding the names or descriptions of the target dataset's classes, *e.g.* "The image shows *<label>*.". For simplicity, we refer to a model trained in a contrastive language-image pre-training setting and fine-tuned or probed for a downstream task as CLIP model. Closely related to CLIP, the ALIGN (Jia et al., 2021) model is a family of multimodal dual encoders that learn to represent images and text in a shared embedding space. Instead of Vision-Transformers (ViT) or ResNet models ALIGN uses the

EfficientNet (Tan & Le, 2019) and BERT (Devlin et al., 2019) models as vision and text encoders. These encoders are trained from scratch on image-text pairs (1.8B pairs) via contrastive learning.

These models and their zero-shot capabilities display significant promise for widely-applicable tasks like image retrieval or search (Radford et al., 2021). For instance, since image and text are encoded in the same representational space, these models can find relevant images in a database given text or relevant text given an image. More importantly, the relative ease of steering CLIP toward various applications with little or no additional data or training unlocks novel applications that were difficult to solve with previous methods, e.g., as we show, inferring the offensiveness in images.

**Carried Knowledge of Pre-trained Large-Scale Models.**   As already described, with training on raw text, large-scale transformer-based language models revolutionized many NLP tasks. Recently, Radford et al. (2021), Ramesh et al. (2021) and Jia et al. (2021) presented encouraging results that a similar breakthrough in computer vision will be possible. Besides the performance improvements in generation, regression, and classification tasks, these large-scale language models show surprisingly strong abilities to recall factual knowledge present in the training data (Petroni et al., 2019). Further, Roberts et al. (2020) showed that large-scale pre-trained language models' capability to store and retrieve knowledge scales with model size.

Since such models are often trained on unfiltered data, the kind of knowledge acquired is not controlled, leading to possibly undesirable behavior such as stereotypical and derogatory associations. However, Schick et al. (2021) demonstrated that these models can recognize, to a considerable degree, their undesirable retained knowledge and the toxicity of the content they produce. The authors further showed that a language model with this ability can perform self-debiasing to reduce its probability of generating offensive text. Furthermore, Jentzsch et al. (2019) and Schramowski et al. (2020) even showed that the retained knowledge of such models carries information about moral norms aligning with the human sense of *"right"* and *"wrong"* expressed in language. Similar to (Schick et al., 2021), Schramowski et al. (2021) demonstrated how to utilize this knowledge to guide autoregressive language models' text generation to prevent their toxic degeneration.

In this work, we investigate if we are able to utilize the carried knowledge of large-scale vision models in a similar way, *i.e.* detecting possible offensive images in large-scale vision datasets.

## 3    Pre-trained models are able to infer offensiveness in images.

Inspired by these previous results, we utilize a pre-trained multi-modal (language-vision) model to investigate its carried visual knowledge. We make use of the multimodality by prompting the model with natural language to analyze if and to which extent it is able to infer the offensiveness in images.

**Offensive images.**   Let us start by defining the term "offending" and describing it in the context of images. According to the Cambridge dictionary[2], "offending" can be phrased as "unwanted, often because unpleasant and causing problems". Additionally, in the context of images and text, according to Law Insider[3]: *Offending Materials means any material, data, images, or information which is (a) in breach of any law, regulation, code of practice or acceptable use policy; or (b) defamatory, false, inaccurate, abusive, indecent, obscene or menacing or otherwise offensive; or (c) in breach of confidence, copyright or other intellectual property rights, privacy or any other right of any third party.* In this work, we focus on images following the definition (b). This definition aligns with definitions of previous work detecting hate speech (Gomez et al., 2020) and offensive product images (Gandhi et al., 2020).

As Gandhi et al. (2020) have described, technical challenges of building such a system are, among others, the lack of adequate training data, an extreme class imbalance, and changing test distribution. However, we will showcase that recent pre-trained models trained on large-scale data guided by natural language supervision are able to distinguish between inappropriate, possible offending image content and other images based on their carried knowledge acquired during the pre-training phase. This is due to the self-supervised learning and the implied task-independence of the involved transformer.

---

[2]https://dictionary.cambridge.org/dictionary/english/offending, accessed on 3rd October 2021
[3]https://www.lawinsider.com/dictionary/offending-materials, accessed on 3rd October 2021

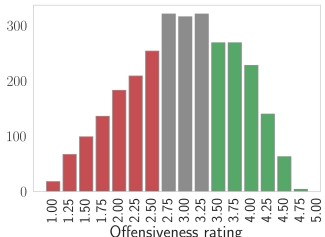

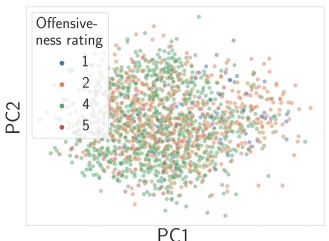

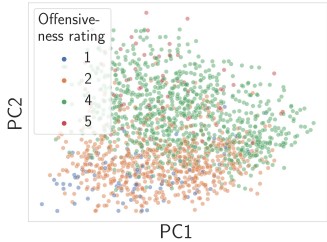

(a) SMID data distribution.

(b) ResNet50 pre-trained on ImageNet1k.

(c) ViT-B/16 pre-trained on WebImageText via CLIP.

Figure 2: The SMID dataset. a) $rating < 2.5$ are samples with possible offensive content and $> 3.5$ images with positive content. b-c) PCA visualization of SMID feature space using different pre-trained models. Coloring of data samples indicates the moral rating of the image's content. A rating of four and five are immoral content and one and two moral content.

Note that offensiveness is a concept that is a function of social norms and people have diverse sentiments. In the present study, we define offensive content based on a data-centric approach. Therefore, the investigated "offensiveness" may primarily surfaces from the group of people that have generated the selected data and the annotators but also the pre-trained model's implicit knowledge. Both, the model and the data could contain biases which can result in ethical issues. We refer to the ethics statement (Section 6) for details.

To steer the model towards detecting (morally) offensive image content, we used the images contained in the Socio-Moral Image Database (SMID) (Crone et al., 2018). The dataset contains 2,941 images. It covers both the morally good and bad poles of a range of content dimensions including objects, symbols as well as actions portrayed. In a large-scale survey, these images were annotated by 2,716 participants. Fig. 2 shows the density distribution of the annotated labels. Based on their findings, Crone et al. (2018) divided the data into *good* (green; mean rating $> 3.5$), *bad* (red; mean rating $< 2.5$), and neutral (grey; rest) images. As the creators suggest, we discretised a rating $< 2.5$ as (morally) offending, and rating $> 3.5$ as not offending, *cf.* Fig. 2. We split the dataset (962 negative images and 712 positive images) into two random splits for our experiments. The training set contains 90%, and the test set the remaining 10% images. In the following experiments, 10-fold cross-validated results are reported.

**Deep Learning to classify offensive image content.** In the context of the Christchurch mosque shooting video streamed on Facebook, officials at Facebook replied that one reason the video was not detected and removed automatically is that artificial intelligence systems are trained with large volumes of similar content. However, in this case, there was not enough comparable content because such attacks are rare[4]. Also Gandhi et al. (2020) have described the lack of adequate training data to train a classifier to, in their case, detect offensive product content. We further investigate this issue with our next experiment by fine-tuning a common pre-trained model on few offensive labeled images.

To measure how well a model can encode what a human could consider to be offending, we considered the above-mentioned Database (in total 1,674 images). We start by training a deep vision model on this dataset. Similar to Gandhi et al. (2020) we chose the ResNet50 architecture (He et al., 2016) pre-trained on ImageNet datasets (Deng et al., 2009). Fig. 2(b) shows a dimension reduction via PCA of the embedded representations of the pre-trained model, *i.e.* before trained on the SMID dataset. Based on this dimension reduction, it is unclear if the ImageNet1k pre-trained ResNet50 variant is able to infer offensiveness in images reliably. Also, after training the network, the performance of the fine-tuned (training all model parameters), as well as linear probed model (*cf.* Tab. 1), shows inconclusive results; even if the performance increases when a larger dataset (ImageNet21k) is used. Specifically, the resulting weak precision and weak recall of probing the ImagNet1k based models shows us that it has issues classifying truly offensive images as well as distinguishing between truly

---

[4]https://www.washingtonpost.com/technology/2019/03/21/facebook-reexamine-how-recently-live-videos-are-flagged-after-christchurch-shooting/, accessed on 4th October

| Arch. | Dataset | Accuracy (%) | Precision | Recall | F1-Score |
|---|---|---|---|---|---|
| ResNet50 | ImageNet1k | $78.36 \pm 1.76$ | $0.75 \pm 0.05$ | $0.74 \pm 0.09$ | $0.76 \pm 0.02$ |
| | | $80.81 \pm 2.95$ | $0.75 \pm 0.02$ | $0.81 \pm 0.02$ | $0.80 \pm 0.03$ |
| | ImageNet21k | $82.11 \pm 1.94$ | $0.78 \pm 0.02$ | $0.80 \pm 0.05$ | $0.78 \pm 0.04$ |
| | | $84.99 \pm 1.95$ | $0.82 \pm 0.01$ | $0.85 \pm 0.06$ | $0.82 \pm 0.04$ |
| | WebImageText | $\circ 90.57 \pm 1.82$ | $\circ 0.91 \pm 0.03$ | $\circ 0.89 \pm 0.01$ | $\circ 0.88 \pm 0.03$ |
| ViT-B/32 | WebImageText | $94.52 \pm 2.10$ | $0.94 \pm 0.04$ | $0.91 \pm 0.02$ | $0.92 \pm 0.01$ |
| ViT-B/16 | WebImageText | $\bullet \mathbf{96.30 \pm 1.09}$ | $\bullet \mathbf{0.95 \pm 0.02}$ | $\bullet \mathbf{0.97 \pm 0.01}$ | $\bullet \mathbf{0.97 \pm 0.02}$ |

Table 1: Performances of pre-trained models ResNet50 and ViT-B. The ResNet50 is pre-trained on ImageNet1k, ImageNet21k (Deng et al., 2009) and the WebTextImage dataset (Radford et al., 2021). The ViT is pre-trained on the WebTextImage dataset. On the ImageNet datasets, we applied linear probing (top) and fine-tuning (bottom), and on the WebImageText-based models, soft-prompt tuning. The overall best results are highlighted **bold** with the • marker and best on the ResNet50 architecture with ∘ markers. Mean values and standard deviations are reported.

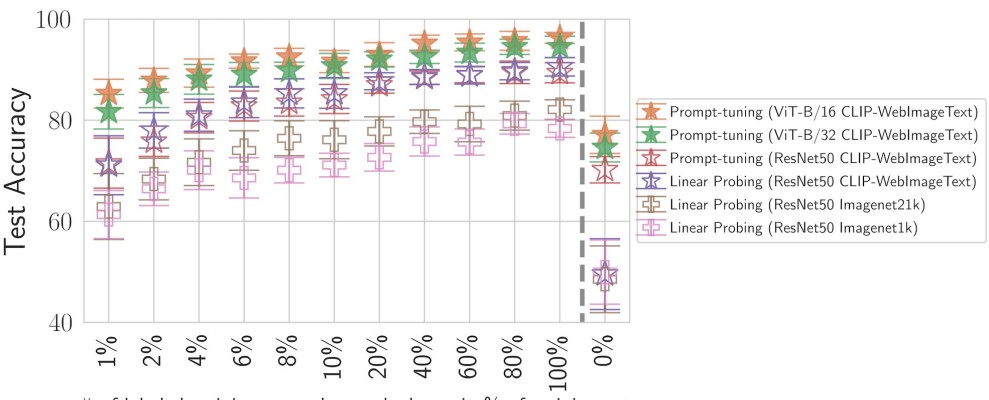

Figure 3: Performances of pre-trained models ResNet50 and ViT-B. The ResNet50 is pre-trained on ImageNet1k, ImageNet21k (Deng et al., 2009) and the WebTextImage dataset (Radford et al., 2021). The ViT is pre-trained on the WebTextImage dataset. On the ImageNet datasets, we applied linear probing (top), and on the WebImageText-based models soft-prompt tuning. Tuning was performed on different sizes of the SMID training set where $100\%$ corresponds to 1506 images. One can see that steering CLIP towards inferring the offensiveness in images requires only little additional data. In contrast to other pre-trained models, it therefore provides a reliable method to detect offending images.

non-offensive and offensive images. After fine-tuning the whole model, it can increase its recall; however, the precision stays comparable low.

This supports the previous findings mentioned above. Next, we will consider these models as baselines to investigate if more advanced PMs trained on larger unfiltered datasets carry knowledge about offensiveness.

**Pre-trained, natural language guided models carry knowledge about offensiveness.** By training on over 400M data samples and with natural language supervision, CLIP (Radford et al., 2021), and other similar models (Jia et al., 2021), acquire (zero-)few-shot capabilities displaying a significant promise for applications with little or no additional data. Next, we will investigate if this includes the detection of offensive image content.

Fig. 2(c) shows the embedded representations of the ViT-B/16 model pre-trained on WebImageText via Contrastive Language-Image Pre-training (Radford et al., 2021). One can observe that the ViT's learned representation encodes knowledge of the underlying task, *i.e.* distinguish offensive and not offensive images without being explicitly trained to do so. These results confirm our assumption that due to the natural language supervision, CLIP implicitly acquired knowledge about what a human could —depending on the context— perceive as offending.

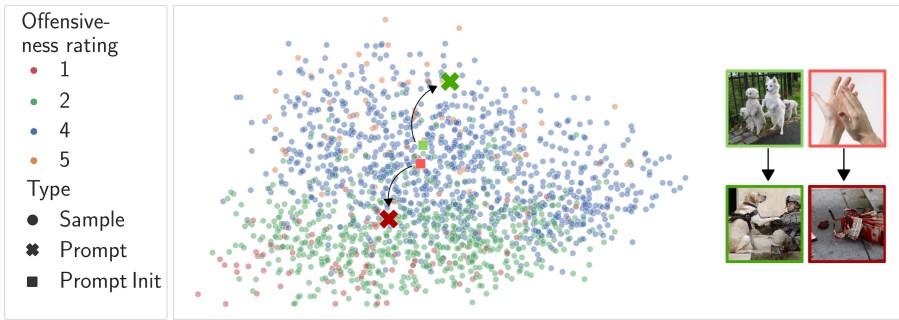

Figure 4: Soft-prompt tuning on vision-language representation space. The squared data samples visualize the locations of the initial prompt and the crosses the final prompts. On the right, the nearest image samples for each prompt are displayed.

Furthermore, the natural language supervision of CLIP allows us to probe the model without training it (zero-shot). More precisely, as with the previous model (ResNet50 pre-trained on ImageNet), the images are encoded via the pre-trained visual encoder. Instead of training a linear classifier, we operate on the similarity of samples, in this case, the cosine similarity, in the representational space:

$$Sim(x, z) = \frac{E_{visual}(x) * E_{text}(z)}{||E_{visual}(x)||_2 * ||E_{text}(z)||_2} \, , \tag{1}$$

where $E_{visual}$ and $E_{text}$ are the visual and text encoders, and $x$ an image sample and $z$ a prompt. We embedded the classes, as suggested by Radford et al. (2021), into natural language prompts such as "This image is about something *<label>*.", which has shown to be a good default helping to specify that the text is about the content of the image. Following the collection of human annotations contained in the SMID dataset (Crone et al., 2018), we applied various prompt classes: *bad/good behavior*, *blameworthy/praiseworthy*, *positive/negative* and *moral/immoral*. Whereby, the labels *positive* and *negative* resulted in the best zero-shot performance.

Fig. 3 (0%) shows that this zero-shot approach utilizing the implicit knowledge of the CLIP models is already performing on par with the ImageNet-based PMs which were fine-tuned on SMID. However, we noticed that the zero-shot approach is able to classify true-negative samples well but performs less well on classifying positives. This suggests that both, or at least the prompt corresponding to the positive class label, are not chosen optimally. The nearest image neighbors extracted from the SMID dataset (*cf.* Fig. 4 top-right) confirm this observation.

**No need to learn new features: Learning how to ask the model.** The previously defined prompts may not be the optimal way to query the model's implicit knowledge to infer the offensiveness in images. To further steer the model, we, therefore, searched for optimal text embeddings, *i.e.* optimize the prompts, which is also called (soft-) prompt-tuning (Zhong et al., 2021; Qin & Eisner, 2021). As an optimization task we defined the distinction of offensive and not offensive images and optimized the prompts by gradient descent as follows:

$$\hat{\mathbf{z}} = \arg\max_{\mathbf{z}}\{L(\mathbf{z})\} \, , \tag{2}$$

where

$$L(\mathbf{z}) = -\frac{1}{|X|} \sum_{\mathbf{x} \in X} \mathbf{y} \, \log(\hat{\mathbf{y}}) \, , \quad \text{with } \hat{\mathbf{y}} = \text{softmax}(Sim(\mathbf{x}, \mathbf{z})) \, . \tag{3}$$

Note, that we do not updated the parameters, $\theta$, of $E_{visual}$ and $E_{text}$. The term $\mathbf{y}$ is the ground truth class label and $X$ a batch during the stochastic gradient descent optimization. The resulting prompts are shown in Fig. 4 and clearly portray on the one side possible offending image content and the other side positive content.

Next, we will evaluate the resulting CLIP model equipped with the newly designed prompts. Fig. 3 shows that even a small portion of the training data (*e.g.* 4%, 60 images) increases the vision transformer's (ViT-B) performance to over 90%. In general, the vision transformer outperforms the pre-trained ResNet50 models. Furthermore, the vision transformer with higher model capacity

outperforms the smaller variant, indicating that not only the dataset's size is important, but also the capacity of the model. Training with the full training set reaches a final test accuracy of $96.30\% \pm 1.09$ (*cf.* Tab. 1). These results clearly show that large-scale pre-trained transformer models are able to infer the offensiveness in images and that they already acquire this required knowledge during their pre-training phase guided by natural language supervision.

## 4 MACHINES ASSIST TO DETECT OFFENDING IMAGES IN CV BENCHMARKS

Next, we utilize the pre-trained CLIP model, and the SMID-based selected prompts to identify possible offending images from popular computer vision benchmark datasets. As Birhane & Prabhu (2021) we focus on ImageNet and use its most-popular subset the ImageNet-ILSVRC-2012 dataset as an example.

Using our previously described approach the pre-selection by CLIP extracts possible offensive images. However, offensiveness is subjective to the user and, importantly, the task at hand. In our case, the steered model may primarily mirror the moral compass and social expectations of the 2,716 annotators. Therefore, it is required that humans and machines interact with each other, and the human user can select the images based on a given setting and requirements. Hence, we do not advise removing specific images but investigate the range of examples and offensiveness selected by the system and thereby document the dataset. We here provide an exemplary list of contents and disguised images (Fig. 5). Additionally, we provide Python notebooks with the corresponding images along with the classifier in the supplemental material. Moreover, to enforce targeting possible strongly offensive images, we determined the prompts by shifting the negative threshold to a rating of 1.5 instead of 2.5.

Due to the complexity of offensive context, we separate the identified offensiveness into offending objects, symbols, and actions in images.

**Objects.** The ImageNet1k dataset, also known as ImageNet-ILSVRC-2012, formed the basis of task-1 of the ImageNet Large Scale Visual Recognition Challenge. Hence, all images display animals or objects. Therefore it is not surprising that the largest portion of offensive content concerns negative associated objects and animals. In total, 40,501 images were identified by the offensiveness classifier, where the objects "gasmask" (797 images), "guillotine" (783), and "revolver" (725) are the top-3 classes. However, whereas most people would assign these objects as morally questionable and offensive, they can not be treated as offensive when training a general object classifier. The same applies to the animal-classes tick (554) and spider (397).

To infer the offensiveness of images contained in ImageNet, it may be more applicable to investigate classes with only a small portion of possible offensive images. Next to injured (*e.g.* "koala", "king penguin") and aggressive animals (*e.g.* "pembroke", "redbone"), our proposed classifier detects caged (*e.g.* "great pyrenees", "cock") and dead animals (*e.g.* "squirrel monkey", "african elephant"). Additionally, objects in inappropriate, possible offensive scenes, like a bathtub tainted with blood ("tub") are extracted.

**Symbols.** Furthermore, one is able to identify offensive symbols and text on objects: several National Socialist symbols especially swastika (*e.g.* "mailbag", "military uniform"), persons in Ku-Klux-Klan uniform (*e.g.* "drum"), insults by *e.g.* showing the middle finger (*e.g.* "miniature pinscher", "gorilla", "lotion"), and inappropriate text such as "child porn" ("file") and "bush=i***t f*** off USA" ("pay-phone").

**Actions.** In addition to objects and symbols, our proposed classifier is able to interpret scenes in images and hence identify offensive actions shown in images. Scenes such as burning buildings (*e.g.* "church") and catastrophic events (*e.g.* "airliner", "trailer truck") are identified. More importantly, offensive actions with humans involved are extracted such as comatose persons (*e.g.* "apple", "brassiere", "tub"), persons involved in an accident (*e.g.* "mountain bike"), the act of hunting animals (*e.g.* "African elephant", "impala"), a terrifying person hiding under a children's crib ("crib"), scenes showing weapons or tools used to harm, torture and kill animals (*e.g.* "hamster") and people (*e.g.* "hatchet", "screwdriver", "ballpoint", "tub").

Furthermore, derogative scenes portraying men and women wearing muzzles ("muzzle"), clearly misogynistic images *e.g.* harmed women wearing an abaya, but also general nudity with exposed

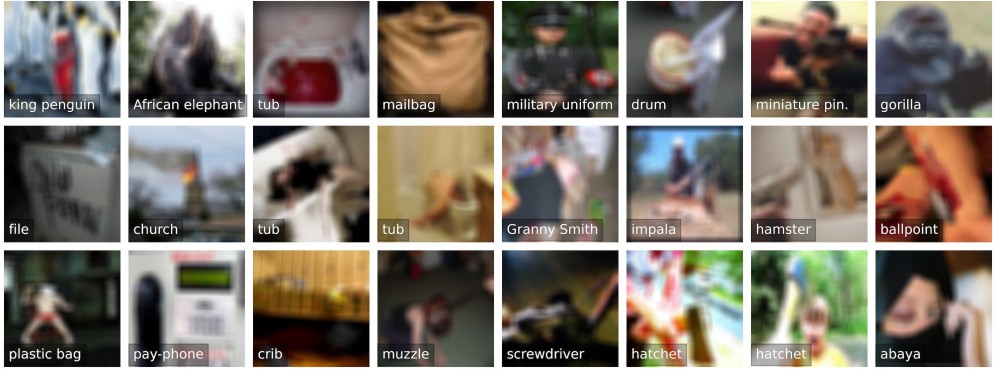

Figure 5: Exemplary hand-picked images with offensive content from the pre-selection of our proposed method. The images visualize the range of offensiveness (objects, symbols, actions) detected. Due to their apparent offensive content, we blurred the images. Their content can be inferred from the main text.

genitals (*e.g.* "bookshop", "bikini", "swimming trunks") and clearly derogative nudity (*e.g.* "plastic bag") are automatically selected by our proposed method. Note that *e.g.* the misogynistic image showing a harmed woman wearing an abaya was not identified by the human hand surveyed image selection of Birhane & Prabhu (2021). Therefore, we strongly advocate utilizing the implicit knowledge of large-scale state-of-the-art models in a human-in-the-loop curation process to not only partly automatize the process but also to reduce the susceptibility to errors.

## 5 CONCLUSION

In recent years, deep learning approaches, especially transfer learning from models pre-trained on large-scale supervised data, have become standard practice for many applications. To train such models, a tremendous amount of data is required. As a result, these datasets are insufficiently filtered collections crawled from the web. Recent studies (Gebru et al., 2018; Birhane & Prabhu, 2021; Bender et al., 2021) have revealed that models trained on such datasets, and the resulting models for downstream tasks benefiting from these pre-trained models, implicitly learn undesirable behavior, e.g., stereotypical associations or negative sentiment towards certain groups. Consequently, there is an urgent need to document datasets and curate their content carefully. Unfortunately, current processes are tedious, error-prone, and do not scale well to large datasets.

To assist humans in the dataset curation process, we, therefore, introduced a novel approach utilizing the implicit knowledge of large-scale pre-trained models and illustrated its benefits. We showed that CLIP (Radford et al., 2021) retains the required knowledge about what a human would consider to be offending during its pre-training phase. As a result, it offers a solution to overcome previous issues, namely the lack of sufficient training data to identify offensive material automatically. In this regard, we have outlined a new solution to assist the curation process on large-scale datasets. On the example of the ImageNet-ILSVRC2012 dataset, we showcased that our proposed approach can identify additional inappropriate content compared to previous studies. Our approach can be transferred to any other vision dataset.

In future work, we thus plan to extend our analysis to other datasets such as the OpenImage dataset (Kuznetsova et al., 2020) and multi-modal datasets (Jia et al., 2021). Further possible avenues for future work are the extensions of the proposed method to multi-label classification to directly separate offensive objects, symbols, and actions or derive other categories of offensive content. Moreover, classifying different levels of offensiveness could further provide details to document datasets; however, this could require additional data. Since the underlying model of our proposed classifier is a deep learning model, it inherits its black-box properties. This makes it hard to understand why the model is identifying specific images. Applying explainable AI methods such as (Chefer et al., 2021) to explain the reasoning process could lead to further improvement of the curation process.

## 6 ETHICS STATEMENT

Our proposed method provides a solution to automatically infer offensiveness in images with the intention to assist the curation process and documentation of datasets. However, we strongly advise applying such methods in a human-in-the-loop setting. Since CLIP models themselves are trained with weak supervision on not freely accessible data sources and only the pre-trained models are provided by its' creators, it is unclear if *e.g.* social biases are inherent to the model. Details about possible biases and other potential misuses (*e.g.* surveillance) of the CLIP models can be found in the original work of Radford et al. (2021).

Furthermore, in the present study, offensive content is detected by a data-centric AI system, where the model is steered towards the sentiment reflected in the data. Therefore, next to the model's contained biases, the investigated "offensiveness" may primarily surfaces from the group of people that have generated the selected data and the annotators but also the pre-trained model's implicit knowledge. An interesting and important avenue for future work is addressing what different groups of society, e.g. different cultures, would consider as offensive by the creation of datasets representing their corresponding sentiments. As other social norms, offensiveness is constantly evolving. This makes it necessary to update the data, system and documentations.

## 7 REPRODUCIBILITY STATEMENT

The code to reproduce the figures and results of this article, including pre-trained models, can be found in the publicly available repository. Furthermore, we provide the source code needed in order to determine prompts to steer the CLIP model towards inferring offensiveness in images and apply it to detect possible offending images in vision datasets. The figures with disguised images are provided in original form in the supplement material.

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
