# OpenReview forum: "Inferring Offensiveness In Images From Natural Language Supervision"
_ICLR.cc/2022/Conference — ICLR 2022 Submitted_

### Official Review · Reviewer_yFoK · 2021-11-01

**Correctness:** 2
**Technical Novelty And Significance:** 1
**Empirical Novelty And Significance:** 2
**Recommendation:** 3
**Confidence:** 4

**Details Of Ethics Concerns:**

I don't think this a strict flag due to some moral failure on the part of the authors, but rather a flag I am opening given the sensitivity of this topic area that I want to call attention to (detecting "offensiveness"). See also my review on the difficulties with the paper's definition/scope of "offensiveness".

**Main Review:**

To begin, I don’t think the work adequately defines what they are scoping/defining by "offensiveness". In Section 3, the work draws from some other works and focus in on “defamatory, false, inaccurate, abusive, indecent, obscene or menacing or otherwise offensive” as their scope. In what follows in the rest of the paper, there is an assumption that a large model trained on large data solves for this definition, when in fact, it’s like using an answer to “how” to a question about “what”. As a result, the problem statement of the paper is rather muddled since there’s essentially a “you know it when you see it” argument to the question of "what is offensive content?". Just to name a few other quandaries with problem scoping: to “whom” are we considering the content to be “offensive”? Since “offensiveness” is a concept that is a function of social norms, over what period of time are we considering or is the modeling a function somehow of time?

These questions have important design implications, for example, in the empirical experiments of this work, they discretize ratings in their Socio-Moral Image Database and threshold on <2.5 and >3.5 as offending and non-offending, respectively, but since it’s not clear how we are defining the “whom” of offensiveness, it’s unclear if these choices make sense [N.B. authors followed prior art with those thresholds, but that in itself is a choice by the authors]. To what extreme do we take “injustice anywhere is a threat to justice everywhere” (Martin Luther King Jr)? If it was found to be offensive to one rater? Do we define it at the level of a locale or people group sharing similar socio-cultural values? These are challenging philosophical questions even that I do not expect an ICLR paper to fully address, but a position on the scope needs to be made explicit, otherwise, the objective is unclear and it's difficult to even assess if this paper is making progress on the problem.

Further, I was confused by some of the reasoning in the presentation, which summarizing using quotes follows as:

> "...training data has been shown to have problematic characteristics resulting in models that encode e.g. stereotypical and derogatory
associations...  large, uncurated, Internet-based datasets encode e.g. dominant and hegemonic views..."

> "...we investigate modern vision PMs trained on large-scale datasets, in particular, the Contrastive Language-Image Pre-trained model (CLIP) (Radford et al., 2021) and argue that they themselves pave a way to mitigate the associated risks... they encode implicit knowledge to infer offensiveness in images overcoming previous issues, namely the lack of adequate and sufficient training data."

> "...we demonstrate that our approach can be utilized to annotate offensive images in vision datasets and, therefore, reliably assist the curation process of such datasets."

> "We illustrate our approach on the popular ImageNet-ILSVRC-2012 dataset and show that large computer vision datasets contain additional inappropriate content, which previous documentations had not detected."

The logic seems be that large models trained on large data have undesirable biases and to address this, they propose to use another large model trained on large data to filter out the undesirable biases because they believe a priori that it, like other such models is attuned to the undesirable biases and can be used to pre-train a supervised task to identify one dimension of these biases: “offensiveness”. There’s a weird “who watches the watchmen” problem here. It’s unclear if there is actual a reinforcing loop where model used for filtering (CLIP) has biases and as a result, any downstream tasks will just narrowly focus on its representing of those biases. I will grant that the authors acknowledge this in the Ethics Statement, but I actually think it’s far more central the methods proposed and there's a need to address it head-on versus just having it as an Ethics addendum.

Ultimately, perhaps most puzzling is that despite the argument that such a filtering technique could reduce harmful biases in large datasets like ImageNet1k, the impact of the qualitative examples provided in Section 4 is opaque. For example, are the examples given coming from a high-recall setting where we want to trigger on even very low softmax scores? Even if these are identified in ImageNet1k, do we have reason to believe the base rates are sufficiently high that a naively trained model would encode these biases? If so, why was this then not validated with experiments?

**Summary Of The Paper:**

This work investigates using CLIP for pre-training a model on the Socio-Moral Image Database (prior art) for the task of “offensiveness” detection for which they argue will assist in reducing bias in large models trained from large corpora like ImageNet.

**Summary Of The Review:**

Overall, the paper puts forward a number of interesting empirical experiments, but because the problem statement (“offensiveness”) to begin with is not well-scoped, it then becomes difficult to assess the contributions of the proposed methodology. In addition, the ultimate impact of the work is only shown qualitatively with example concepts (and occurrences) that it’s identified in ImageNet1k, but the results of these are not validated and its impact on say models trained on ImageNet1k without these “offensive” examples is not quantified.

---

> ### Author Response · Authors · 2021-11-17
> **Thanks for your time and efforts. Below please find our responses to your concerns point by point.**
>
> **Inadequate definition of "offensiveness".**
> As described in your review, ``offensiveness'' is not easy to define since "it is a concept that is a function of social norms". We fully agree on this and also state this in our introduction: "It remains an
> open question how to infer general offensiveness in images, including abusive, indecent, obscene, or menacing content [...]".
>
> Therefore, after describing the general definition of "offensiveness", we introduce the selected dataset we use for our data-centric approach, i.e. we define offensiveness by examples, i.e., images labelled by 2,716 annotators. These images visualize possible (morally) offending actions, symbols or objects (cf. Crone et al.).
> Therefore, we would kindly disagree with your comment that it is not an adequate definition. However, we agree that this could be described in more detail and will revise our manuscript accordingly.
>
> **To "whom" are we considering the content to be "offensive"? Over what period of time are we considering?**
> These two questions can also be answered based on our data-centric approach's dataset. The dataset is labelled by 2,716 annotators. Since we steer the CLIP model with the contained images, the model may primarily ("primarily" since we utilize its implicit knowledge) mirror the annotators' sentiment. The same applies to the period of time. However, here, we also need to take the group of people (authors of the dataset) that have selected the data into account.
> We will stress this more in our manuscript and will update it accordingly.
>
> Nevertheless, our present work shows that we can utilize and steer CLIP's implicit knowledge to classify offensive images. As we described in the ImageNet evaluation and our conclusion, the content to be considered offensive has to be defined based on the task and involved stakeholders. In our case, the dataset annotators' sentiment defines the offensiveness.
>
>
> **Impact of the qualitative examples is opaque. Do we have reason to believe the base rates are sufficiently high that a naively trained model would encode these biases?**
> This seems like a fundamental question: Do we first train a model (on potentially unclean) and determine their biases, or do we carefully create datasets to prevent possible biases beforehand? In our work, we follow the arguments of Gebru et al., which align with the World Economic Forum Global Future Council on Human Rights (2018): "How
> to Prevent Discriminatory Outcomes in Machine Learning". The forum suggests that all entities should document the provenance,
> creation, and use of machine learning datasets in order to avoid discriminatory outcomes. See also "Datasheets for Datasets" by Gebru et al. (2018).
>
> In the present work, we demonstrate the documentation process based on ImageNet and explicitly state that we do not filter the dataset, rather use ImageNet as a popular example to illustrate problematic images contained in Machine Learning datasets. Cf. Section 4: "Hence, we do not advise removing specific images but investigate the range of examples and offensiveness selected by the system and thereby document the dataset."
>
> It is clear that a dataset should not contain images, e.g. showing violence against women. These kinds of images can be identified by our approach since these concepts are contained in the dataset our data-centric approach is based on. Similar to previous work, we do not find it necessary to measure the impact of these images, at least in the scope of our work.
> However, it is a very interesting question, and we will add this to our conclusion to motivate further research.
>
> **Addressing the ethical issues more prominent than an Ethical statement.**
> Thank you for the feedback. Since ICLR provides a special location for ethical statements, we followed that guideline. However, we agree that addressing ethical issues more prominent is important and will update our manuscript accordingly by describing the ethical issues more intensive and more prominent in the main text.

---

> > ### Comment · Reviewer_yFoK · 2021-11-23
> > **Thanks for the comments. Some follow-up...**
> >
> > Thank you for the time and effort in responding.
> >
> > I emphasize again that this an incredibly challenging problem area. My hope with the original review was to solicit clarity on the proposal to make this problem more tractable. I appreciate the spirit of what the work seeks to accomplish as well as its cautionary ethos, but it’s the execution that I remain less convinced about.
> >
> > Two main concrete areas that I remain unconvinced:
> > * **Impact of “data-centric” definition of “offensiveness”**: I agree that there is a need to have humans in-the-loop for the scoping of what is “offensive”, but I’m not as convinced with the approach of watering down human judgments to a one-hot by a simple threshold. The authors acknowledge the complexity of defining “offensiveness,” but the model and pipeline need to reflect this complexity. For example, I might have been less skeptical if the authors emphasized characteristics like wide variance in inter-rater agreement on “offensiveness” in the dataset and then showed how incorporating we could incorporate this variation in a model design, e.g. by sample weighting or selection.
> > * **”Representation” vs “use”**: Similar to discussion in the interpretability community, just because a concept is represented by a model, it does not mean that it is necessarily used w.r.t. the target task. For example, a model could be representing concept up through some layer L so that in the L+1 layer it can wash out the concept’s effects so that the downstream task is robust to the bias. To be sure, aiming to document the provenance to a finer degree is admirable, but this is a necessary yet not sufficient contribution. It’s critical to show the effects having created greater data transparency, otherwise it is transparency for transparency’s sake.

---

### Official Review · Reviewer_zJbR · 2021-11-02

**Correctness:** 3
**Technical Novelty And Significance:** 2
**Empirical Novelty And Significance:** 3
**Recommendation:** 3
**Confidence:** 3

**Details Of Ethics Concerns:**

This paper studies inferring offensiveness in scraped web images which are widely used by modern computer vision models, and the authors propose to reveal the offensiveness from a potentially biased model (CLIP or ALIGN, both of which are used as pre-trained checkpoint without further analysis on the data being used for training). More experiments and analysis is needed to verify whether other bias/fairness issues are involved in this process.

**Main Review:**

Strength:
1. This paper focused on an important aspect of fairness/dataset bias in modern deep learning models, and showed an interesting direction to reveal questionable instances from the dataset. As the authors advocate this approach if combined to the human-in-the-loop protocol for dataset collection, may help creating more responsible datasets while saving some tedious human efforts.
2. The authors give very detailed introduction, background and related work sections, highlighting the importance of the task.
3. The code and pre-trained checkpoint provided in the supplementary might be useful for future work.

Weakness:
1. The technical contribution of this paper is very limited. Although it's arguably as important to identify new question and explore new application of existing techniques, essentially what this paper did is just an extension of CLIP-based models. The baseline experiments are simply finetuning/linearly probing existing models (ResNet-50 and ViT) on a public dataset (SMID), where the offensiveness is more of a synonym of "immoral/blameworthy". The proposed work regarding CLIP is more like a specific zero-shot learning application (i.e. offensiveness), and the prompt tuning is also highly related to previous work. Maybe this work is more suitable for a dedicated conference/workshop rather than a generic top-tier AI conference like ICLR.
2. One important section that is missing from this paper is the analysis of precision/recall of the proposed work. In Table 1, the authors show precision/recall/F1 but this is on SMID dataset (if I'm understanding correctly), where the "ground truth" labels are inferred based on discretization of rating (<2.5 or >3.5). This is different from the true offensiveness detection. Specifically, since the authors already compared with Birhane & Prabhu (2021) with their hand surveyed image selection, how accurate does the proposed model match the performance of the human selection? In Page 9 the author mention that one image of Figure 5 (misogynistic image) is missed by Birhane & Prabhu, but I'm curious in a statistical level how well do they agree. Alternatively a user study might be useful, that within the 40,501 detected images how many are truly "offensive"? Right now it seems to me that the authors only show several methods effective on SMID dataset, but the key contribution of "inferring offensiveness" is built upon the assumption that SMID rating is a proxy for offensiveness.
3. The concerns about ethical issues are not fully addressed. See Ethics Concerns section for more comments.
4. Some minor issues need to be resolved: For example, Figure 4 caption is it "On the right" (not "On the left")? Footnote 4 on Page 5, "Oktober" -> "October". The format of citation is mistaken in multiple occurrences, such as Figure 3 caption "(Deng et al. 2009)", and second paragraph of page 6, in "CLIP (Radford et al. 2021)", "other similar models (Jia et al 2021)".

**Summary Of The Paper:**

This paper described an interesting idea to leverage massively pre-trained models (specifically CLIP-based models) to infer offensiveness in images. The authors first gave detailed literature review regarding a recently raising concern about inappropriate images in computer vision datasets, and also performed several finetuning/probing baseline experiments to illustrate the feasibility to detect such inappropriate contents. Next the authors explored the possibility to mitigate the potential risk by utilizing the implicit knowledge learned by CLIP-based pre-
trained models. Lastly the authors choose ImageNet for a proof-of-concept validation and show that the proposed approach can discover previously neglected offensive images.

**Summary Of The Review:**

In general I think it's a solid paper with proper experiments as proof-of-concept, but the lack of technical contribution and missing the actual analysis on the detection of offensive contents are major weakness of the submission. For top-tier AI conference like ICLR this paper is not good enough.

---

> ### Author Response · Authors · 2021-11-17
> **Thanks for your time and efforts. Below please find our responses to your concerns point by point. (1)**
>
> We appreciate the positive evaluation of the relevance of our work for the fairness in AI community.
>
> **Contribution not suitable for ICLR.**
> Thank you for agreeing on the importance of identifying new questions and exploring new applications of existing techniques. However, we have to admit that we are a bit confused that the reviewer further comments that our contribution is too limited for ICLR. Machine Learning and AI have matured so much that many people like Andrew Ng even consider moving from models to data as the key to making further progress again.
> That is we got to make the data work as well [Link](https://www.forbes.com/sites/gilpress/2021/06/16/andrew-ng-launches-a-campaign-for-data-centric-ai/?sh=2434ebc074f5). The documentation and selection are crucial to prevent discriminatory outcomes in machine learning, cf. World Economic Forum Global Future Council on Human Rights (2018).
> Even ICLR itself states the importance of societal considerations of representation learning in its Call for Papers.
>
> **Missing analysis of precision/recall.**
> Thank you for the feedback. This is correct; we do not provide many details on the precision/recall metrics results, which may leave the reader unaware of key issues of the naively trained models. We will update our manuscript accordingly with the following extension:
>
> In addition to accuracy, we provide precision and recall metrics.
> The resulting weak precision and weak recall of probing the ImagNet1k based models shows us that it has issues classifying truly offensive images as well as distinguishing between truly non-offensive and offensive images. After fine-tuning the whole model, it can increase its recall; however, the precision stays comparable low.
>
> On the contrary, the CLIP model provides a reliable (high recall and high precision) approach without the need of fine-tuning the model to infer offensiveness.
>
> **Contribution of "inferring offensiveness" is built upon the assumption that SMID rating is a proxy for offensiveness.**
> The SMID dataset contains clearly (morally) offensive images. As also reviewer **yFoK** states offensiveness "is a concept that is a function of social norms". The SMID dataset represents what the 2,716 annotators would consider offensive, and therefore our data-centric AI approach primarily mirrors their moral compass and social expectations. We indeed assume during the documentation of ImageNet using our approach (Section 4) that these 2,716 annotators represent our stakeholders. We acknowledge that this should be described more clearly and will update it accordingly.
>
> However, as one can see, based on the extracted images, this is a fairly reasonable assumption. To name an example, it should be clear that showing actions like harming women should not be contained in Machine Learning datasets to prevent discriminatory outcomes.
>
> **How accurate does the proposed model match the performance of the human selection?**
> This is a very interesting question.
> The referenced human selection of Birhane et al. investigates images showing persons including the following categories: "beach-voyeur-photography", "exposed-private-parts", "verifiably pornographic", and "upskirt". In ImageNet's training set 59 images are identified, where 30 are categorized as "beach-voyeur-photography", 10 as "exposed-private-parts", 3 as "verifiably pornographic" and 11 as "upskirt".
> Since the dataset we used to define offensiveness does not include nudity (nevertheless, it could identify some images showing nudity), it only identifies 4 (2 "exposed-private-parts" and 2 "upskirt") of them as (morally) offensive. However, our approach identifies further misogynistic, e.g. expressing violence against women, which is not included in the study of Birhane et al. (2021).
>
> Therefore, we stated in our original manuscript (Introduction, second last paragraph): "With our study we therefore strongly advocate for curating and documenting a dataset by the categories and models provided by Birhane & Prabhu (2021) but also by taking the possible general offensiveness in images into account."
>
> Nevertheless, we agree that especially "verifiably pornographic" and "exposed-private-parts" are generally offensive, and we will run experiments including such examples.
> In the first preliminary experiments, we used half of the images to steer CLIP. The resulting model is able to identify the hold-out images as offensive, indicating that we can indeed also steer the model to detect "verifiably pornographic" and "exposed-private-parts".
> Furthermore, this nicely shows that users can define which images should be considered as offensive depending on their preferences and requirements without much effort (with a few examples).
>
> We will run further experiments and investigate what additional ImageNet images are detected by the model. We will keep you posted.

---

> > ### Author Response · Authors · 2021-11-17
> > **Further comments (2)**
> >
> > **User study.**
> > We agree with the reviewer that a user study could improve the presentation of the results. However, we do not find it necessary to illustrate the need for careful dataset creation and documentation (via transformers).
> > It is clear that specific areas of the described range of images (Section 4) should not be contained in Machine Learning datasets to prevent discriminatory outcomes.
> >
> > **Ethical issues are not fully addressed.**
> > Thank you for the feedback.
> > We address the potential ethical issues in our ethical statement and refer to the original work of CLIP for a detailed evaluation of the potential biases. However, as reviewer **yFoK** suggests, highlighting the ethical issues in the main text is of importance, and we will update our manuscript accordingly.
> >
> > **Minor issue.**
> > Thank you for pointing out these minor issues. We revised our manuscript accordingly.

---

> > > ### Comment · Reviewer_zJbR · 2021-11-17
> > > **Any rebuttal about the technical contribution of this paper?**
> > >
> > > I would like to thank the authors for the clarification.
> > >
> > > One comment I think the authors may have misinterpreted is Weakness 1 in my original review. Please note that the recommendation for submitting to a dedicated workshop is not based on the *topic* but on the *limited technical contribution*. In fact from the initial reviews, this concern seems to be shared by all reviewers (score for technical novelty is 1, 2, 2), and I don't see the authors giving a very convincing rebuttal on that. Again, my first criticism is not **Contribution not suitable for ICLR**, but that **technical contribution is neither novel nor exciting**.
> > >
> > > Please feel free to emphasize the *technical* contribution of this paper in your response if you believe we missed something. No need to repeat the big picture though.
> > >
> > > Thanks.

---

> > > > ### Author Response · Authors · 2021-11-18
> > > > **Thank you for attending the discussion.**
> > > >
> > > > We did not misinterpret the reviewer's stated Weakness 1 but may have chosen incorrect wording. Please accept our apologies for that.
> > > >
> > > > However, we are still confused since the reviewer argues that the lack of technical contribution is an important reason for rejection at ICLR (see summary of review). First, as ICLR's call for papers and previous studies (like the last year's publication of the ETHICS dataset by Hendrycks et al. "Aligning AI With Shared Human Values ", ICLR 2021, https://openreview.net/forum?id=dNy_RKzJacY) clearly show ICLR is not restricted to just novel technical contributions. Even if a reviewer was concerned that the ICLR format is not suitable, the acceptance of such papers at ICLR shows the opposite.
> > > >
> > > > Moreover, we argue that creating a system (adapting state-of-the-art representation learning techniques) that can reliably infer offensive image content is a novel and significant technical contribution. While the way we interconnect and prompt the systems is already novel, the creation of the system per se is even more important. Showing that transformers can actually help anticipate ethical issues in visual data sets is a key step to align representation learning with our society. "Offensive" is not yet another label. It is a highly complex concept, and it is an exciting and, in turn, important message that transformers are able to capture it. As Yan LeCun famously said, "ML systems are biased when data is biased". Our approach helps to reduce this form of bias encoded in data.

---

> > > > > ### Comment · Reviewer_zJbR · 2021-11-18
> > > > > **Thanks for the update.**
> > > > >
> > > > > Thanks for reminding us again how important the problem is. I strongly agree this paper presents a very good story (as I stated in the original review).
> > > > > I don't have any other follow-up comments.

---

### Official Review · Reviewer_dcH1 · 2021-11-08

**Correctness:** 3
**Technical Novelty And Significance:** 2
**Empirical Novelty And Significance:** 2
**Recommendation:** 3
**Confidence:** 3

**Main Review:**

The contribution of the paper is not wholely clear to this reviewer.

First a short summary of the paper: (1) the paper picks a fairly obscure dataset for prediction (2) constructs a good classifier using prompt fine tuning and CLIP (cool!) (3) applies it to ImageNet and provides a few categories with images deemed immoral.

Overall, the paper is pitched as a dataset analysis tool for problematic images. But little to no effort is put into presenting results of analysis, beyond listing a few categories in ImageNet that had a lot of 'immoral' judgements predicted. I am left essentially knowing that a classifier can be built to predict morality based on an obscure dataset.

On a technical level, while the results of prompt tuning of CLIP in particular could be new, the application is a fairly obvious combination of existing ideas, although I am excited to know its possible. I would rather have seen a full exploration of prompt fine tuning in CLIP rather than a narrow application to this morality dataset.

**Summary Of The Paper:**

The paper constructs a classifier of whether or not an image is offensive. This is operationalized by finding a dataset from the psychology community of a few thousand images and ordinal judgements of 'morality' from a study. Prediction of these judgements is predictably hard, in no small part because the data is small. To combat these issues, the paper uses CLIP and soft-prompt tuning. Soft-prompt tuning appears to be very effective for CLIP in this context reaching accuracies of over 95%, where baseline fine tuning only gets about 85%.

**Summary Of The Review:**

The paper does not provide deep analysis of offensive content in any visual dataset. Technically, the paper combines currently trendy and exciting techniques in fairly straightforward ways to create a classifier from little data for morality judgements. This technical contribution is not fully explored, and in itself does not appear novel.

---

> ### Author Response · Authors · 2021-11-17
> **Thanks for your time and efforts. Below please find our responses to your concerns point by point.**
>
> **Unclear contribution.**
> We appreciate that the reviewer finds our technical contribution interesting. Thanks! However, we must admit that we are also confused about the reviewers' argument of unclear contribution.
> We present for the first time the important insight that transformers can help anticipate ethical issues in datasets!  It is not our interest to highlight prompt-tuning in combination with models such as CLIP. Even if it is quite unexplored in the context of vision representation learners guided by natural language, one can find several studies in the NLP domain. We can transfer these approaches since we investigate a multimodal (text-image) model.
>
> More importantly, as Reviewer **zJbR** describes our study focus on the critical aspect of fairness in AI, which is highlighted as a relevant topic in ICRL's Call for Papers.
> We clearly describe the motivation of the present study as well as our contribution to data documentation and curation.
>
> To summarise, we tackle the urgent need to carefully document datasets and curate their content. Therefore, to assist humans in the dataset curation process, we introduced a novel approach and illustrated the approach's benefits. Our approach offers a solution to overcome previous issues and identifies offensive material automatically.
> Furthermore, as Reviewer **yFoK** states, it is non-trivial to define what makes an image offensive. Therefore, we propose a data-centric AI approach, leading us to the reviewer's following comment.
>
> **Obscure dataset.**
> To be honest, we are a bit offended by the review's comment calling the published SMID dataset of Crone et al. (PLOS ONE, 2018) obscure. First of all, PLOS ONE itself is a well-known journal. Moreover, the authors of the dataset are well cited and have other publications, e.g., in the Proceedings of the National Academy of Sciences (PNAS, impact factor > 10)) among others. Therefore, we would like to ask the reviewer to be a bit more detailed in this regard. The SMID dataset contains 2,941 images labelled by 2,716 participants. As mentioned beforehand, we chose a data-centric AI approach. It allows us to steer the model -- with its implicit knowledge acquired during pre-training -- to classify what people consider to be (morally) offending. Since the SMID dataset's images show a broad range of possible offensive content, including symbols, objects, and actions, it is indeed suitable for the task we aim for. Therefore, we are puzzled why the reviewer describes the dataset as ``obscure'', questioning its suitableness, particularly given that our results clearly show the usefulness of the dataset for the task at hand.
>
>
> **Narrow application.**
> As described in the previous comments, we disagree that the present study is a narrow application.
> The World Economic Forum *Global Future Council on Human Rights* (2018) states the importance of documenting the provenance, creation, and use of Machine Learning datasets in order to "prevent discriminatory outcomes in machine learning". In our opinion, world-leading machine learning conferences like ICLR can be frontiers in addressing this emerging problem.
> Therefore investigations such as Birhane et al. (WACV 2021) and Bender et al. (FAccT 2021), as well as datasets such as Hendrycks et al. (ICLR 2021) and methodological contributions like ours are of importance for AI conferences such as ICLR.
>
> **Presenting results of analysis.**
> We present the results of our study by first evaluating our approach quantitatively on the hold-out test set of the SMID dataset and secondly by applying it to the popular large-scale vision dataset ImageNet. The former shows that our data-centric approach outperforms previous approaches and is a reliable (precision/recall) method to detect offensive content. Based on one of the most popular vision datasets, the latter shows that indeed inappropriate images, here (morally) offensive images, are contained in ImageNet and can be documented by our approach. With our documentation in Section 4, we, however, clarify that offensiveness depends on the task and involved stakeholders, e.g. our data-centric approach steers the CLIP model based on the moral compass and social expectations of 2,716 users.
> It is clear that specific areas of the described range of images should not be contained in Machine Learning datasets to prevent discriminatory outcomes. In any case, we agree that an additional presentation next to the textual description could strengthen the present study. However, we disagree that a deeper analysis is necessary to illustrate the need for careful dataset creation and documentation.
>
> We think that these experiments sufficiently show our significant contribution to documenting large-scale vision datasets. This task becomes even more relevant in the light of Apple's initiative against the spread of Child Sexual Abuse Material (CSAM) online. The EU Commission is considering taking similar but different actions as well.

---

### Author Response · Authors · 2021-11-17
**We thank all reviewers for their feedback. However, we also have to admit that the reviews seem to be rather harsh. Let us try to explain this while addressing the main issues raised.**

We make an important contribution: we show that transformers can actually help to anticipate ethical issues in visual training sets. This is a challenging problem since "offensive" is not yet another label. It is a highly complex concept, and it is exciting that transformers are able to capture it. Moreover, this is of high interest to the ICLR community, as it has a genuine interest in training their models on appropriate datasets. This may also explain why "societal considerations of representation learning including fairness, safety, privacy, and interpretability" is listed as relevant topic on the call for papers.

---

### Decision · Program_Chairs · 2022-01-20

**Decision:**

Reject

**Comment:**

The authors set out on an important question of whether abstract and culturally specific concepts like offensiveness can be detected in images. The novelty of this work comes in part from tackling this question and attempting to create a technology which can operationalize it.  However, despite the authors' insistence that offensiveness is not "just another label", in practice the work treats it very much that way and therefore does not present a compelling innovation either in modeling or in juxtaposition to other labeling tasks.  Known training and inspection techniques are used on existing representations and more powerful models with more training data generalize better. It is unclear what is novel in the approach or unique to offensiveness over other labels (including abstract ones).